# A Review of Registered Randomized Controlled Trials for the Prevention of Obesity in Infancy

**DOI:** 10.3390/ijerph18052444

**Published:** 2021-03-02

**Authors:** Seema Mihrshahi, Danielle Jawad, Louise Richards, Kylie E. Hunter, Mahalakshmi Ekambareshwar, Anna Lene Seidler, Louise A. Baur

**Affiliations:** 1Department of Health Systems and Populations, Faculty of Medicine, Health and Human Sciences, Macquarie University, Sydney, NSW 2109, Australia; 2NHMRC Centre of Research Excellence in the Early Prevention of Obesity in Childhood Sydney School of Public Health, The University of Sydney, Sydney, NSW 2109, Australia; kylie.hunter@ctc.usyd.edu.au (K.E.H.); mahalakshmi.ekambareshwar@sydney.edu.au (M.E.); lene.seidler@ctc.usyd.edu.au (A.L.S.); louise.baur@health.nsw.gov.au (L.A.B.); 3Sydney School of Public Health, Faculty of Medicine and Health, The University of Sydney, Sydney, NSW 2006, Australia; daniellejawadd@gmail.com (D.J.); louise.a.richards@gmail.com (L.R.); 4NHMRC Clinical Trials Centre, The University of Sydney, Locked bag 77, Camperdown, NSW 1450, Australia; 5Sydney Medical School, Faculty of Medicine and Health, The University of Sydney, Sydney, NSW 2006, Australia

**Keywords:** behaviours, childhood, infant feeding, interventions, obesity, prevention, physical activity

## Abstract

Childhood overweight and obesity is a worldwide public health issue. Our objective was to describe planned, ongoing and completed randomized controlled trials (RCTs) designed for the prevention of obesity in early childhood. Two databases (World Health Organization International Clinical Trials Registry Platform, ClinicalTrials.gov) were searched to identify RCTs with the primary aim of preventing childhood obesity and at least one outcome related to child weight. Interventions needed to start in the first two years of childhood or earlier, continue for at least 6 months postnatally, include a component related to lifestyle or behaviours, and have a follow up time of at least 2 years. We identified 29 unique RCTs, implemented since 2008, with most being undertaken in high income countries. Interventions ranged from advice on diet, activity, sleep, emotion regulation, and parenting education through to individual home visits, clinic-based consultations, or group education sessions. Eleven trials published data on child weight-related outcomes to date, though most were not sufficiently powered to detect significant effects. Many trials detected improvements in practices such as breastfeeding, screen time, and physical activity in the intervention groups compared to the control groups. Further follow-up of ongoing trials is needed to assess longer-term effects.

## 1. Introduction

Childhood obesity is a worldwide public health issue with an estimated 38.3 million children under five years affected by overweight or obesity in 2019 [1]. Between 1980 and 2015, the global prevalence of obesity in children in the 2–4 year age group rose almost twofold, from 3.9 to 7.2% in boys and from 3.7 to 6.4% in girls [2]. Children who are overweight in early childhood are more likely to still be affected by overweight or obesity in later childhood, adolescence, and adulthood [3], and obesity in childhood can affect a child’s immediate health as well as their educational attainment and quality of life [4]. Obesity has been associated with an increased risk of non-communicable diseases such as type 2 diabetes, cardiovascular disease, and many cancers [5]. This in turn has implications for health systems as well as economies.

With recognition of the rise in prevalence of obesity in early life and its resulting consequences, there has been an increasing focus on the first 1000 days, from conception to 2 years, as a critical life stage to prevent obesity [4,6]. Obesity-related behaviours such as poor diet quality, decreased physical activity, increased sedentary behaviours, and decreased sleep duration are established in, and track from, early life [7]. Evidence has accumulated about the best ways to support parents to establish protective behaviours and influence their child’s trajectory toward healthy growth, and several early intervention trials have been conducted. In Australasia, for example, four RCTs of early obesity prevention interventions including over 2000 mother-child dyads have been undertaken since 2008 [8,9,10,11] and the results have been combined in an individual participant data prospective meta-analysis [12] undertaken by the Early Prevention of Obesity in Childhood (EPOCH) collaboration [13]. The interventions that were tested included a combination of promoting and extending the duration of breastfeeding, introducing appropriate healthy solid foods after 6 months, limiting discretionary foods, promoting parental responsiveness to feeding cues, ensuring adequate sleep and activity patterns, and limiting screen time. The prospective meta-analysis showed that children in the intervention group had lower body mass index (BMI) z-scores at 18 to 24 months than children in the control group (−0.12 adjusted mean; 95% confidence interval, −0.22 to −0.02, *p* =0.017), which is equivalent to a 2% absolute reduction in the prevalence of overweight and obesity. Improvements were also detected in behaviours that may protect against obesity, such as reduced television viewing time, improved feeding practices, and increased breastfeeding duration. Although the effect size of the interventions on BMI z-score was modest, it is important on a population-level scale, and the observed improvements in behaviours in early life may have consequences in later childhood. 

Several trials evaluating interventions for obesity prevention in early life started in the last decade. However, intervention content and delivery features within these trials are often not clearly defined, and therefore interventions are difficult to replicate. In addition, because of the complexity of most interventions, it is not clear which components of the intervention contributed to the positive effects, if any, of the interventions. It also remains unknown at which age to commence interventions, and what are the optimal duration and intensity of the interventions and the best delivery methods and agents for effective implementation. Important contextual factors that underlie interventions such as level of background health care and features of the target population may influence the level of effectiveness [14]. Qualitative analyses including process analysis of behavioural interventions showing how they work and in which population groups are lacking [15]. 

Clinical trial registries offer a valuable resource to understand the landscape of planned and ongoing clinical trials in a particular area [16]. Since the International Committee of Medical Journal Editors (ICMJE) declared clinical trial registration an “ethical obligation” in 2005, registration rates of trials have increased substantially, providing a more complete database of clinical trials that are planned, ongoing, or recently completed [17,18]. The Australian NHMRC Centre of Research Excellence in the Early Prevention of Obesity in Childhood (EPOCH CRE) has established a repository of registered trials to identify and describe the features of early obesity prevention intervention trials and identify evidence gaps in this critical area [19]. Trial registries are regularly searched to identify RCTs evaluating preventive behavioural interventions designed to reduce childhood obesity according to predefined criteria. The ultimate aims of this repository are to improve data quality by sharing information about tools and measurements and to promote collaboration among trialists. 

In this review we aim to conduct descriptive analyses to understand features of registered RCTs that aim to reduce the risk of overweight and obesity in children in the first two years. We review the intervention components, the features of delivery and implementation including mechanisms and agents, the theoretical models that they are based on, and the target populations, and we examine the funding sources. Our purpose is to summarise the accumulating evidence for the primary prevention of obesity in children and identify promising intervention features in order to inform future health promotion programmes and policies.

## 2. Methods

### 2.1. Search Strategy 

As part of our ongoing research work in the EPOCH CRE, the World Health Organisation (WHO) International Clinical Trials Registry Platform and ClinicalTrials.gov were searched every three to five months since April 2016 by K.E.H. in order to identify eligible trials to add to the EPOCH CRE Trial Repository [19]. Search terms included variations of “infant”, “child” “overweight”, “obesity”, and “prevention”.

Registration records were screened and studies were included if (1) they were RCTs, (2) the main aim was to prevent childhood obesity, (3) the intervention commenced within the first two years or antenatally, (4) they continued for at least 6 months postnatally, (5) they included a component related to lifestyle, and (6) the trial had a follow-up period of at least two years from baseline. Pilot studies were excluded. Published papers for each of the eligible trials were identified using the registration number and/or study title via the PubMed database. For trials with multiple publications, all relevant publications were used to extract pertinent information. For studies which had published results, information was extracted from the final published papers, while studies with only a protocol or clinical trial registration record were coded as per the latest protocol or record. The latter studies were included, as our purpose was to describe not only completed interventions and their results, but also the types of interventions that are currently being planned or undertaken.

### 2.2. Data Extraction

For each eligible trial the following information was extracted: registration number, study title, principal investigator, protocol/results publication year, recruitment country/ies, study design, number of participants, timing of intervention commencement, timing of baseline data collection, primary outcome(s), secondary outcome(s), delivery agent, main intervention components, and the type of control group. 

Detailed data on intervention characteristics were extracted using an adapted version of the Template for Intervention Description and Replication (TIDieR) reporting guidelines [20]. Intervention commencement was coded as antenatal or post-natal and intervention setting was coded as clinic/community-based, home-based, or both. Delivery agent was coded according to who provided the intervention to participants (e.g., nurses, dietitians, psychologists, etc.) and intervention mode was coded as individual, group, individual and group, telephone or mobile application, or a combination of these. Intervention delivery referred to the implementation materials (such as educational handouts, educational videos) as well as procedures (such as home visits or through mobile applications). Target populations were coded by ethnicity, socio-economic position, literacy level, and parental weight status. Funding sources were categorised as government, non-government organisation, university, industry, or mixed. Where available, theoretical model, cost data, and biomarkers were also extracted. Where data for some variables were missing, trialists were contacted to provide further information.

Data were extracted by two investigators (D.J. and L.R.) and then cross-checked for accuracy by a third investigator (S.M.). Inconsistencies were settled through discussion. 

### 2.3. Quality Assessment

All trials with published weight-related data were included in the quality assessment using the Cochrane Risk of Bias tool version 2 (RoB2) for randomised trials [21]. Risk of bias assessment was undertaken on all publications including follow up publications. Each trial was assessed independently by two reviewers (S.M. and D.J.) as “low risk” or “high risk” of bias or having “some concerns”. Disagreements were resolved through discussion. No trials were excluded from the review based on the results of the risk of bias assessment. 

### 2.4. Data Synthesis

Findings are presented in data tables and summarised in the text. Only descriptive comparisons were performed, as described in Section 2.2 above.

## 3. Results

Electronic searches identified 2292 records. After removing duplicates and applying the inclusion and exclusion criteria, 29 unique trials met our eligibility criteria and are summarised in Table 1 (for more complete information see also Appendix A). To date 38% (n = 11) have published outcomes [22,23,24,25,26,27,28,29,30,31,32]. Of the trials that have not yet published outcomes, 12 are ongoing, one has not progressed because of funding issues, and the remaining 5 trials have a status that is unclear in terms of their level of progression/completion. Most trials were conducted/planned in high income countries, with the majority being undertaken in the USA (n = 12, 41%), with the remainder being in Australia (n = 5, 17%), the Netherlands (n = 3, 10%), Sweden (n = 2, 7%), New Zealand (n = 2, 7%), Italy (n = 1, 3%), and Spain (n = 1, 3%). Three trials (10%) were planned in low- and middle- income countries (Mexico, Guatemala, and China), although one has not progressed due to lack of funding [33].

The majority of trials are parallel group RCTs (n = 20, 69%). There are three trials with factorial RCT designs, which have more than one intervention arm, and there are six cluster RCTs for which participants were randomised at the group level. 

It is worth noting that three trials—INFANT Extend [43], CHAT [48], and Greenlight Plus [57]—build on the earlier foundation trials of INFANT [9], Healthy Beginnings [8], and Greenlight [38], respectively. There are similarities to the foundation trials in the intervention components, but the durations of interventions are lengthened [43], or new methods of delivery are being trialled [48,57].

### 3.1. Intervention Characteristics

In approximately one third of the 29 trials (n = 9, 31%) the interventions have commenced or plan to commence antenatally, and in a further eight trials (28%) intervention commencement occurred when the child was one month old or younger. In the remaining 12 trials (41%) the interventions commenced when the child was between 1 and 12 months of age. The duration of interventions ranged from nine months to six years, with an average duration of 23.3 months (SD 14.8). In three trials, the duration of the intervention was unclear (and there was no response from trial authors who were contacted). A total of 83% (n = 24) of trials targeted “parents” or “mothers and caregivers”, while 17% (n = 5) specifically targeted mothers.

Most interventions (55% n = 16) were delivered or plan to be delivered in community settings such as clinics, a further six (21%) were based within the home, and seven interventions (24%) used a combination of home and community settings.

In terms of intervention delivery mode, the majority of interventions (n = 10, 34%) were delivered face-to-face individually, four (14%) were delivered in a group setting, such as a parent support group, and a further four (14%) used a combination of individual and group delivery. In two trials (7%) the interventions were completely delivered via telephone or mobile application, and the remaining trials used a combination of delivery modes for the interventions (n = 9, 31%).

Most trials used multiple forms of delivery methods for the interventions (Table 2). In terms of materials, the majority used some form of educational handouts (n = 24, 83%) and were either delivered at the home visit, clinic or group session, or mailed out. Two trials (7%) involved educational videos, and five trials (17%) included an educational website or application. Three trials used an “educational tool kit” tailored to low literacy populations. In terms of the procedures, six trials (21%) involved consultations with health professionals over the phone, and in four trials (14%) the intervention was delivered through text messages. For more information about each individual trial, please refer to Appendix A.

Interventions were delivered by a range of health professionals and researchers (Table 2), with nurses being the most common (n = 9, 31%). Five interventions were nurse-led in clinics and four were delivered through nurse home visits. Dietitians delivered the intervention in five trials (17%), community health workers also in five trials (17%), trained research assistants in four (14%), and paediatric residents/paediatricians also in four (14%). Other delivery agents included lactation consultants, general practitioners, midwives, physiotherapists, and trained sleep specialists. In six trials (21%) a combination of delivery agents was used. For more information about delivery agents for each individual trial, please refer to Appendix A.

### 3.2. Intervention Components/Content

Most trials included between three and eight intervention components and target multiple behaviours as part of the interventions (Table 3). The most common component was providing general advice about healthy dietary behaviours in children (n = 24, 83%), followed by encouraging play and activity (n = 20, 69%), breast/bottle feeding advice (n = 16, 55%), and targeted parenting practices, especially education around hunger and satiety cues (n = 13, 45%). Several trials targeted sleep promotion (n = 13, 34%), parental modelling of behaviour (n = 13, 45%), or limiting small-screen and TV time (n = 9, 31%). Other intervention components included how and when to introduce solids (n = 10, 34%), limiting junk foods (n = 6, 21%), and how to deal with fussy eating (n = 5, 17%). Only one trial included information about growth monitoring and growth charts as part of the intervention. For more information about intervention components and key messages for each individual trial, please refer to Appendix A. 

### 3.3. Target Populations

Fifty-two percent of trials (n = 15) targeted a locally representative population, while the remainder targeted either those of low socio-economic position (n = 7, 24%), ethnic minorities (n = 5, 17%), a low literacy population (n = 1), or a population of parents affected by overweight/obesity (n = 1, 3%). It should be noted that the five studies that targeted ethnic minorities could also be grouped as low socio-economic status populations, but they were coded to the ethnic minority group as this was the main focus of the study [44,47,49,51,54]. 

### 3.4. Theoretic Basis of Trials

Of all trials, the use of a theoretic model or theory was stated in 12 (41%), and many of these (n = 7, 24%) used multiple theories (Appendix A). The most commonly used theories were social cognitive theory (n = 9, 31%) and social learning theory (n = 5, 17%). Three trials (10%) used the Health Belief Model. 

### 3.5. Funding Sources 

A total of six trials (21%) reported solely government funding. Of the remaining trials, the majority (n = 14, 48%) reported some form of governmental funding in addition to other funding sources. Most of these were jointly funded by non-governmental organisations, universities, and industry. The majority of US studies (n = 9, 75%) were funded by the National Institutes of Health and The National Institute of Diabetes and Digestive and Kidney Disease, while the Australian and NZ studies were funded by the Australian National Health and Medical Research Council and the Health Research Council NZ, respectively (Appendix A). Universities co-funded (n = 10) 34% of studies. 

While no studies were solely funded by industry, 6 (21%) had some industry funding, with Meat and Livestock Australia, Heinz, Danone Nutricia, and Karitane Products Society contributing to three studies (10%). Two Latin American studies, the SPOON studies in Mexico [33] and Guatemala [53], were to be partially funded by the PepsiCo Foundation, but in recent communications one trial had been cancelled [33]. In addition to providing funding, Danone Nutricia provided jars of baby food and printed information materials for participants in the Baby’s First Bites trial [52].

### 3.6. Cost Effectiveness Data 

Only five (17%) of the trials stated in the protocols or other publications that they planned to collect (or have already collected) data on costs of the intervention or conduct cost effectiveness analyses. The Healthy Beginnings trial completed a retrospective economic evaluation, where intervention resources were determined from local health district records [58]. Healthcare utilisation was determined using patient level data linkage, and it was estimated that the program could be delivered for just over AUD$700 per child with a cost-effectiveness ratio of AUD$376 per 0.1 reduction in BMI z-score, which is regarded as a moderately priced intervention.

Three trials (10%) indicated that they will conduct detailed cost effectiveness analyses [48,59,60]. In addition, the PRIMROSE trial plans to conduct a cost utility analysis [59]. The INFANT Extend trial [43] plans to conduct an economic analysis and monitor use of health services in both control and intervention groups to assess whether the program reduces parent’s health seeking behaviours elsewhere.

### 3.7. Biomarkers 

Of the 29 trials, only two planned to collect biomarkers, and of these, neither published the results. The INSIGHT trial [61] conducted genetic testing for appetite, growth, and temperament on the child, mother, and father. The SCHeLTI [56] trial will collect blood from the infant as well as other samples (saliva, stool), and blood and other samples will also be collected from the mother and father. It is unclear which tests will be conducted, but the samples will form part of a biobank and be kept for at least 20 years.

### 3.8. Weight Related Outcomes

To date, 11 trials have reported weight-related outcomes (Table 4), while nine have published protocol papers only [38,42,43,48,49,51,52,57], and for nine information is only available from the clinical trial registration record [33,39,44,45,50,53,54,55,56]. 

A comparison of the reported weight-related outcomes showed heterogeneity in reported measures, with BMI, BMI z-scores, weight-for-length percentiles, waist circumference, and prevalence of overweight and obesity being reported. In addition, the outcomes were reported at different time points (Table 4).

Of the 11 trials with published outcomes, five had follow-up weight outcomes at later time points [34,35,36,37,41], up to age 5 years from baseline, providing further insight into the duration of intervention effects. Retention rates at the first follow up where outcomes were reported ranged from 73% to 92%.

Of the 11 trials, three—POI.nz [26], Healthy Beginnings [22], and INSIGHT [28]—demonstrated statistically significant differences in weight outcomes between the intervention and control groups at the first follow up. Healthy Beginnings demonstrated a significantly lower BMI at 2 years in its intervention group when compared with the control (mean difference −0.29 (95% CI 0.02 to 0.55), *p* = 0.04) [22]; however, there was no statistically significant reduction in BMI observed in follow up at 3.5 or 5 years of age [34]. INSIGHT reported outcomes at 1 and 3 years and demonstrated a reduced weight for length percentile at 1 year of age and a reduced BMI z-score at 3 years of age (mean difference −0.28 (95% CI −0.53 to 0.01), *p* = 0.04) in the intervention group compared to the control group [41]. An overall group effect for the prevalence of obesity at 2 years was observed in the POI.nz study (*p* = 0.027). Participants receiving the “sleep intervention” (including the sleep and combination group) demonstrated a lower prevalence of obesity when compared to the “food, physical activity, and breastfeeding” (FAB) and control groups (OR= 0.54, 95% CI 0.35–0.82) [26]. Children who received the sleep intervention (sleep and combination groups) had significantly lower BMI *z*-scores at age 3.5 years (−0.24; 95% CI: −0.38, −0.10) and at age 5 years (−0.23; 95% CI: −0.38, −0.07) than children who did not (control and FAB groups) [37].

It is important to note that many of the studies were not adequately powered to achieve statistical significance for weight-related outcomes. The effect sizes were generally small, and BMI ranged from mean differences of 0.06 to 0.3 kg/m^2^, and for BMI z-scores differences ranged from 0.01 to 0.3.

### 3.9. Secondary Outcomes

Secondary outcomes are shown in Appendix A. They were related to behaviour change and included duration of breastfeeding, child diet and eating habits, physical activity, screen time, sleep, and health-related parenting practices. Of the 11 trials with published outcomes, we identified a total of 105 unique secondary outcomes collected across the trials, with 75% (n = 81) related to diet and infant feeding. 

### 3.10. Risk of Bias Assessment

Risk of bias assessment was undertaken on the fifteen publications (of 11 trials) with weight-related outcomes. Ten of the fifteen were judged as low risk in all risk of bias domains (67%), a further four were judged as high risk, and one study was judged as having “some concerns”. Almost all (14/15) had a low risk of bias arising from the randomisation process, and one study was deemed as high risk. Two of fifteen studies had some concerns or a high risk of bias arising from “the effect of assignments to interventions” and “measurements of outcomes”, while the remaining thirteen in both domains were low risk. Most studies were judged as low risk in regard to missing outcome data (13/15; 87%); however, the remaining two studies were deemed as high risk of bias in this domain. For selection of reported results, all 15 studies had low risk of bias in this domain. Assessments by trial are shown in Appendix A shows the summary of risk of bias assessments. 

## 4. Discussion

This review examined RCTs with a behavioural/ lifestyle interventions focused on obesity prevention in infancy and shows the range of trials and intervention components being implemented in different contexts at varying stages of completion. We summarised the key characteristics and features of interventions including the behavioural targets, delivery mechanisms and agents, duration of interventions, and target populations. We also examined the theoretical basis for interventions and whether economic evaluations were carried out and assessed the funding sources of these interventions. Interventions were designed to influence a range of important behavioural targets including early feeding and diet, physical activity, sleep, sedentary time, and parenting.

Of the 11 trials that were completed and reported weight-related outcomes, two have shown a small but significant beneficial effect of interventions on child BMI z-score at 2 years of age [22,41], and one found significant improvements in the prevalence of obesity, but not BMI [26]. It is possible that some trials may not have been powered to detect intervention effects for weight-related outcomes, so it remains uncertain whether these interventions are effective in reducing BMI z-score. The EPOCH collaboration demonstrated how combining trial data in a meta-analysis can substantially increase the statistical power to detect an intervention effect for weight-related outcomes [13]. The four included trials had minimal power on their own (all less than 0.35) to detect the observed intervention effect of 0.12 on BMI z-score at *p* < 0.05. However, their combined power was 0.83 [62].

Encouragingly, many trials showed impacts on weight-related behaviours such as improving the duration of breastfeeding, improving healthy food intake, and reducing discretionary foods. These behaviours may be important for long term obesity risk and other health outcomes. This also demonstrates the need to understand the intervention components that are responsible for the changes in behaviours, how they work, and for whom, so that they can be implemented in the most efficient manner in the most appropriate populations [63]. 

It is also important to note that to date only one trial, INSIGHT [41], has shown a sustained effect on BMI lasting until 3 years with all others showing shorter term effects. This phenomenon, described as the “fade out effect” [64], is common in interventions that begin in early childhood and are delivered for short time frames, resulting in the effects not being sustained. This may imply that early interventions need to be of a longer duration and may need complementary interventions as children grow in order for impacts to remain and make substantial changes to a child’s growth trajectory. Trials which have begun recently with duration of implementation of the interventions over 3 and up to 6 years will be able to contribute more definitive answers on whether early interventions can contribute to preventing obesity in the longer term [42,56].

Our review has shown a high degree of heterogeneity in the way primary and secondary outcomes are collected and reported. For example, there was a wide range of weight-related outcomes such as BMI, BMI Z-score, weight for length percentile, waist circumference, and prevalence of overweight and obesity that were assessed at differing ages and time points. Likewise, for dietary outcomes there was a wide range of different measures for similar outcomes. For example, intake of fruit and vegetables was reported as grams/day, times/day and serving size/day which varies with age and between countries. These variations precluded our ability to pool data and conduct a meta-analysis. This highlights the need for standardisation of outcomes related to infant weight and behaviours to facilitate outcome harmonisation and synthesis [65], and the need for a core outcome set for early childhood obesity interventions [66].

Our review included interventions focused on individual behaviours rather than targeting the wider environmental determinants such as the food and built environments, despite many of the interventions being delivered in a community setting. These interventions are important, given the age group—most infants spend a large proportion of their time at home with their parents. Although we are not able to draw specific conclusions given the small sample sizes and lack of power to show effects in most trials, the home setting was used in some studies and may be more advantageous in terms of dose, tailoring, delivery, and participant convenience. Individual or face-to-face interventions may have some advantages over group or indirect methods, such as online interventions [67]. In general, with face-to-face interactions, the intervention delivery agents are more tuned to the individual’s needs and capabilities and can tailor the intervention to suit [68]. During the first year, parents are likely to seek extra advice and therefore they may be more receptive to skill development and parental advice promoting healthy family eating and physical activity [9]. As children learn from and model parental physical and health-related activity levels, targeting parental engagement in infancy also features heavily in recent studies. 

The rise in interventions that use online modes of delivery and delivery through telephone or text messages is encouraging as it means that interventions can be delivered cheaply, quickly, and conveniently at scale. The question remains whether these interventions are effective and as acceptable as those delivered face-to-face individually or in group settings. Process and impact evaluation of these modes of delivery are currently lacking [15]; however, with situations such as the coronavirus disease (COVID-19) pandemic, these modes may be a pragmatic way for interventions to be delivered [69], and future studies should focus on the effectiveness of alternative modes of delivery.

The studies were coded to seventeen domains of intervention content which targeted multiple areas, demonstrating the complexity in the evaluation of trials. In most studies there is a lack of detail about the specific content used within interventions, and this limits the transferability of the study approach. This highlights the importance of deconstructing interventions to their smallest common elements [63] to determine which components are actually driving the effects. In addition, it was difficult to ascertain dose of intervention delivered with few studies providing statistics and information on the average number of clinics/groups/home visits attended and adherence to the protocol, which again may influence the effectiveness of study results. Although some interventions were based on theory and some on multiple theoretical models, most did not state the theoretic basis for the intervention.

It is logical to conclude that by improving health-related behaviours, the flow-on effect will be to improve weight-related outcomes, but as highlighted, many trials did not achieve statistical significance, and this may have been because of a lack of power to show effects. An alternative explanation may be that most of the interventions were delivered for a relatively short duration over one or two years. Longer durations may be important to create sustained change and prevent intervention “fade out”. Interventions in this age group were focused at the individual level of diet and activity and as children get older it would be important to look at the wider social and environmental factors which may play an additional role in the development of obesity.

There is a critical knowledge gap with respect to the ideal duration of interventions and when to intervene. Accumulating evidence has highlighted the influence of the preconception and perinatal periods for preventing childhood obesity and non-communicable disease later in life [70,71,72,73]. It is possible that many trials have not targeted women early enough, with only three trials with published outcomes starting antenatally. Only one trial (which is currently in progress) has randomised women preconception [56]. Interventions starting in preconception could prove more effective, and at least one study found a dose response association between preconception BMI and offspring’s childhood BMI, so future research in this area is warranted [71].

## 5. Limitations

This review was limited by the fact that we only searched trial registries to identify eligible trials. This may mean that some unregistered trials were not identified; however, registration has been a requirement for all trials since 2005 [18], and a recent study found high rates of registration since this requirement came into effect [17]. Another limitation was that for some of the trials, the intervention characteristics were coded from clinical trial registry information, and in some cases, these were very brief and not up-to-date. This may have led to some missing information. We attempted to contact trialists for missing information, and where possible published protocol papers were used to code and categorise the intervention components.

Effectiveness in all of the trials was defined as a statically significant difference in weight-related outcomes in favour of the intervention compared to controls as described by the trial authors. Because some trials had small sample sizes, this would have resulted in inadequate power to detect a significant effect. This problem was overcome in the four Australasian trials, which collaborated to conduct an individual participant data prospective meta-analysis, resulting in improved power to detect effects [13].

In many of the trials, there were high loss to follow up rates. This suggests that the intensity of interventions and participant burden should be considered when designing interventions and health promotion programs. It also shows that some form of early process analysis signifying participant satisfaction with trials should be considered [15].

There were a relatively small number of trials with published data that met the review criteria, and many of the studies included multi-component interventions that made comparisons between trials difficult. However, the number of studies that will progress to reporting outcomes in the future holds promise for more definitive evidence for effective intervention strategies.

## 6. Conclusions

This review shows the breadth of work that is occurring globally across trials for the prevention of obesity in early childhood. We described the key characteristics and features of trials including the behavioural targets, delivery mechanisms and agents, duration of interventions, and key target populations. In the coming years, more trials are likely to publish their results and it will be possible to ascertain which intervention strategies are most effective for prevention of childhood obesity. The complexity of multicomponent interventions means that evaluating these interventions is difficult, and complex methods will need to be employed to show which intervention components at which doses and via which delivery methods are most effective. This may be achieved by bringing together researchers from all relevant trials to share data and learnings to transform the thinking and practices around early childhood obesity prevention—the TOPCHILD collaboration aims to achieve this [74].

## Figures and Tables

**Table 1 ijerph-18-02444-t001:** Characteristics of early intervention studies for the prevention of obesity in infancy (n = 29).

	Registration No	Trial Name/Acronym	Author, Year	Country	Study Design	Number Randomized	Intervention Commencement	Duration of Follow Up
1.	ACTRN12607000168459	Healthy Beginnings	Wen, 2012 [22] Wen, 2015 [34]	Australia	RCT	N = 667	Antenatally	Birth until 5 years
2.	ISRCTN81847050	InFANT	Campbell, 2013 [23] Hesketh, 2020 [35]	Australia	Cluster RCT	N = 542	Mean 3.8 months	4 months until 5 years
3.	ACTRN12608000056392	NOURISH	Daniels, 2013 [24] Daniels, 2015 [36]	Australia	RCT	N = 698	Child age 4 months	4 months until 5 years
4.	NCT00756626	FYCS	Bonuck, 2013 [25]	USA	RCT	N = 300	12 months of age	12 months until 24 months
5.	NCT00892983	POI.nz	Taylor, 2016 [26] Taylor, 2018 [37]	New Zealand	2 × 2 factorial RCT	N = 802	Antenatally	Birth until 5 years
6.	NTR1831	BeeBOFT	van Grieken, 2017 [27]	Netherlands	Cluster RCT	N = 2102	1 month of age	1 month to 36 months
7.	NCT01040897	GREENLIGHT	Sanders, 2014 [38]	USA	RCT	N = 865	Child age 2 months	5 months to 2 years
8.	NCT03370445	Health Literacy and Numeracy	Cruzatt, 2017 [39]	USA	RCT	N = 450 *	Child age 2 months	2 months to 5 years
9.	NCT01167270	INSIGHT	Savage, 2016 [28] Adams, 2018 [40] Paul, 2018 [41]	USA	RCT	N = 279	Child age 1–2 weeks	1–2 weeks to 3 years
10.	NCT01198847	Early STOPP	Sobko, 2011 [42]	Sweden	RCT	N = 200	Child age 1 year	1 year to 6 years
11.	ACTRN12611000386932	INFANT Extend	Campbell, 2016 [43]	Australia	Cluster RCT	N = 540	Child age 3 months	3 months to 36 months
12.	NCT01541761	Starting Early Obesity Prevention Program	Messito, 2017 [44]	USA	RCT	N = 533	Antenatally	Birth to 3 years
13.	NCT01649115	HLPP	Reddy, 2012 [45]	USA	RCT	N = 150	Antenatally	Birth to 5 years
14.	ACTRN12612001133820	Baby-led introduction to solids (BLISS)	Taylor, 2017 [29]	New Zealand	RCT	N = 206	Antenatally	Birth to 24 months
15.	NCT01905072	Preventing Childhood Obesity through Early Guidance	Reifsnider, 2013 [46] Reifsnider, 2018 [47]	USA	RCT	N = 140	Antenatally	1 week to 3 years
16.	ISRCTN16991919	PRIMROSE	Doring, 2016 [30]	Sweden	Cluster RCT	N = 1369	Child age 9–10 months	9–10 months to 4 years
17.	PMC4442409	Early Obesity Prevention	Schroeder, 2015 [31]	USA	Cluster RCT	N = 232	Paediatric visit at 1 month of age	1 months to 5 years
18.	ACTRN12616001470482	CHAT	Wen, 2017 [48]	Australia	RCT (3 arm)	N = 1056	Antenatally	Birth to 1 year
19.	NCT03077425	CHALO	Karasz, 2018 [49]	USA	RCT	N = 360	Home visits at 6 months of age	6 months to 18 months
20.	NCT03131284	Prevention of Obesity in Toddlers (PROBIT) Trial	Morandi, 2019 [32]	Italy	RCT	N = 529	First 2 weeks of life	Newborn to 2 years of age
21.	NL6727 (NTR6938)	Samen Happie!	Karssen, 2017 [50]	Netherlands	RCT	N = 300 *	7–11 months	7–11 months until 4 years
22.	NCT03334266	Family Spirit Nurture, Prenatal–18 Months	Ingalls, 2019 [51]	USA	RCT	N = 338 *	Antenatally	Birth to 24 months
23.	NCT03348176	Baby’s First Bites	Van der Veek, 2019 [52]	Netherlands	RCT factorial	N = 240	Child age 4 months	4 months until 36 months
24.	NCT03399617	SPOON: Guatemala	Gonzalez-Acero, 2018 [53]	Guatemala	RCT	N = 1500 *	0–6 months	0–6 months until 24 months
25.	NCT03438721	Strong Futures	Beck, 2018 [54]	USA	RCT	N = 240 *	Child age 2 weeks	2 weeks until 24 months
26.	NCT03444415	PROGESPI	Perez-Lopez, 2018 [55]	Spain	Cluster RCT	N = 414 *	Antenatally	Birth to 24 months age
27.	ChiCTR1800017773	SCHeLTI	Wu, 2018 [56]	China	Cluster RCT	N = 4000 *	6 weeks	6 weeks until 5 years
28.	NCT03752762	SPOON: Mexico	Martinez, 2018 [33]	Mexico	RCT	N = 1200 *	0–6 months	0–6 months until 24 months
29.	NCT04042467	Greenlight plus study	Rothman, 2019 [57]	USA	RCT	N = 900	First newborn clinic visit	Newborn to 24 months

* estimated number.

**Table 2 ijerph-18-02444-t002:** Intervention delivery materials, procedures, and agents used in early intervention studies for the prevention of obesity in infancy (N = 29). **(Refer to Appendix A for more detail on individual trials)**.

Intervention Delivery		N	% of Total
**Materials**			
	Educational handout	15	52
	Educational handout (image-based)	4	14
	Educational video	2	7
	Low literacy educational tool kit	3	10
	Educational website/app	5	17
	Educational material mailed out	5	17
	Feeding supplement	2	7
**Procedures**			
	Nutrition and parenting support groups	7	24
	Phone call consultation	6	21
	Home visits	12	41
	Educational text messages	4	14
**Agents**			
	Nurse (via home visits)	5	17
	Nurse (via clinic visits)	5	17
	Registered dietitian	5	17
	Lactation consultants	3	10
	Trained research assistants	4	13
	Community health worker	1	3
	Community health worker (via home visits)	4	13
	Nutrition expert	2	7
	Paediatric residents/paediatrician	4	13
	Psychologist	1	3
	General practitioners	1	3
	Midwives	1	3
	Physiotherapist	1	3
	Trained sleep specialists	1	3
	Multidisciplinary team	2	7

**Table 3 ijerph-18-02444-t003:** Intervention components/key messages and advice used in early intervention studies for the prevention of obesity in infancy (N = 29). **(Refer to Appendix A for more detail on individual trials)**.

Advice/Key Message/Component	N	%
Breastfeeding/bottle feeding advice	16	55
Introduction of solids	10	34
Limit junk foods (e.g., sweets)	6	21
Repeat food exposure	3	10
Healthy dietary behaviours in children	24	83
Food serving size	5	17
Parenting/hunger satiety cues	13	45
Parent modelling	13	45
Fussy eating	5	17
Soothe/sleep	3	10
Sleep promotion	10	34
Play/activity	20	69
Tummy time	3	10
TV/screen time	9	31
Oral hygiene practices	1	3
Growth chart education	1	3
Health information technology access education	1	3
Health-communication curriculum	2	7

**Table 4 ijerph-18-02444-t004:** Effect of trial interventions on weight outcomes in early intervention studies for the prevention of obesity in infancy.

Study, Author, Year	Sample Size	Primary Outcome	Reported Outcome at End of Follow Up				Effect Size
			Control Group	Intervention Group			
Healthy Beginnings, Wen, 2012 [22]	N = 667 * N = 497	BMI at 2 years	16.82	**16.53**			Mean difference −0.29 (95% CI, 0.02 to 0.55), *p* = 0.04
Wen, 2015 [34]	* N = 415	BMI at 3.5 years	16.8	16.74			Mean difference −0.06 (95% CI, −0.41 to 0.28), *p* = 0.33
		BMI z score at 3.5 years	0.97	0.89			Mean difference −0.08 (95% CI, −30 to 0.16), *p* = 0.44
	* N = 369	BMI 5 years	16.28	16.31			Mean difference 0.3 (95% CI, −0.30 to 0.37), *p* = 0.06
		BMI z score at 5 years	0.63	0.65			Mean difference 0.02 (95% CI, −0.19 to 0.22), *p* = 0.06
INFANT, Campbell, 2013 [23]	N = 542 * N = 457	BMI (z score) at age 20 months	0.8	0.8			Mean difference −0.01 (95% CI, −0.16 to 0.13), *p* = 0.86
INFANT, Hesketh, 2020 [35]	* N = 361	BMI (z score) at age 3.6 years	-	-			Mean difference 0.05 (95% CI, −0.1 to 0.19)
		Waist circumference at 3.6 years	-	-			Mean difference −0.01 (95% CI, −0.12 to 0.19)
	* N = 337	BMI (z score) at age 5 years	-	-			Mean difference −0.02 (95% CI, −0.2 to 0.16)
		Waist circumference at 5 years	-	-			Mean difference 0.01 (95% CI, −0.17 to 0.20)
NOURISH, Daniels, 2013 [24]	N = 698 * N= 530	BMI z score at age 2 years	0.75	0.61			Mean difference −0.14, *p* = 0.10
Daniels, 2015 [36]	* N= 424	BMI z score at age 5 years	0.41	0.34			Mean difference 0.07, *p* = 0.06
Feeding Young Children Study, Bonuck, 2014 [25]	N = 300 * N = 135	Reduction in weight for length >85th percentile	-	-			OR 1.01 (95% CI, 0.9 to 1.1), *p* = 0.8
POI.nz, Taylor, 2017 [26]		BMI at 2 years	16.9	FAB 17.1	Sleep 16.8	Both 16.8	*p* = 0.086 (overall)
		BMI z score at age 2 years	0.77	FAB 0.92	Sleep 0.68	Both 0.72	*p* = 0.104 (overall)
		Waist circumference at 2 years	46.7	FAB 47.0	Sleep 46.6	Both 46.9	*p* = 0.610 (overall)
		Prevalence of overweight and obesity at 2 years	68	FAB 73	Sleep 61	Both 70	*p* = 0.770 (overall)
		Prevalence of obesity at 2 years	33	FAB 40	Sleep 19	Both 21	*p* = 0.027 (overall) FAB vs. sleep group (odds ratio (OR) 0.46, 95% confidence interval (CI), 0.25–0.83), *p* = 0.011 FAB vs. combination group (OR 0.51, 95% CI, 0.28–0.90), *p* = 0.022 Sleep and combination vs. FAB and control (OR 0.54, 95% CI, 0.35–0.82), *p* = 0.004
POI.nz, Taylor, 2018 [37]	N = 808 * N = 616	BMI z score at age 3.5 years	0.68	FAB 0.81	Sleep 0.54	Both 0.56	*p* = −0.004 (overall) FAB vs. Control difference 0.15 (95% CI, −0.04 to 0.34) Sleep vs. Control difference −0.16 (95% CI, −0.36 to 0.04) Both vs. Control difference −0.18 (95% CI, −0.37 to 0.02)
	* N = 557	BMI z score at age 5 years	0.39	FAB 0.66	Sleep 0.31	Both 0.44	*p* = 0.004 (overall) FAB vs. Control difference 0.25 (95% CI, 0.04 to 0.47) Sleep vs. Control difference −0.14 (95% CI, −0.36 to 0.09) Both vs. Control difference 0.06 (95% CI, −0.29 to 0.16)
BeeBOFT, van Grieken, 2017 [27]	N = 2102 * N = 1543	BMI (mean) at 36 months	15.66	15.78			Mean difference 0.12, *p* = 0.12
		Prevalence of overweight/obesity (%) at 36 months	3.99%	4.77%			0.78% difference in prevalence, *p* = 0.51
INSIGHT Savage, 2016 [28]	N = 291 * N = 250	Weight-for-length percentile at 1 year of age	64.4%	57.5%			Mean difference 6.9%, 95% CI (52.6%−69.0%), *p* = 0.4
INSIGHT Paul, 2018 [41]	* N = 232	BMI z-score at 3 years of age	0.15	−0.13			Mean difference −0.28 (95% CI, −0.53 to 0.01), *p* = 0.04
		Mean BMI percentiles	54th	47th			Difference 6.9 percentile points (95% CI, −14.5 to 0.6), *p* = 0.7
BLISS, Taylor, 2017 [29]	N = 206 * N= 178	BMI z score age 12 months	0.20	0.44			Adjusted difference, 0.21, (95% CI, −0.7 to 0.48)
	* N = 166	BMI z score age 24 months	0.24	0.39			Adjusted difference 0.16, (95% CI, −0.13 to 0.45)
PRIMROSE, Doring, 2016 [30]	N = 1369 * N = 1148	BMI at 4 years of age	16.1	16.0			Mean change −0.11 (95% CI, −0.31 to 0.08), *p* = 0.26
		Waist circumference (cm) at 4 years of age	53	52.5			Mean change −0.48 (CI, −0.99 to 0.04), *p*= 0.07
		Prevalence of overweight and obesity at 4 years of age	15.5%	14.8%			RR 0.95 (95% CI, 0.69 to 1.32) *p* = 0.78
Early Obesity Prevention, Schroeder, 2015 [31]	N = 292 * N = 278	BMI at baseline	15.03	15.29			Mean difference 0.26
		BMI Z score at baseline	−0.152	−0.283			Mean difference −0.435
		Weight at baseline	4.56	4.91			Mean difference 0.35, *p* < 0.006
	* N = 218	BMI at 12 months	17.29	17.23			Mean difference −0.06
		BMI Z score at 12 months	0.539	0.492			Mean difference −0.047
		Weight at 12 months	9.81	9.85			Mean difference 0.04, *p* > 0.05
	* N = 232	BMI at 24 months	16.20	16.34			Mean difference 0.14
		BMI Z score at 24 months	0.218	0.339			Mean difference 0.121
		Weight at 24 months	12.61	12.76			Mean difference 0.15, *p* > 0.05
PROBIT, Morandi, 2019 [32]	N = 569 * N = 529	Prevalence of overweight/obesity (%) at 2 years of age	26.3%	23.8%			3% difference in prevalence, *p* = 0.49

* Number that completed the study. FAB: feeding activity breastfeeding.

## Data Availability

Data is available at EPOCH-CRE. Early Prevention of Obesity Childhood (EPOCH) Trial Repository. https://www.earlychildhoodobesity.com/trial_registry.html (accessed on 24 February 2020).

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
