# Peer review of "A Review of Registered Randomized Controlled Trials for the Prevention of Obesity in Infancy"

_ijerph, 2021, doi:10.3390/ijerph18052444_

Round 1

Reviewer 1 Report

In General: it's a good paper

Title: the title properly explain the purpose and objective of the article

Abstract: abstract contains an appropriate summary for the article, language used in the abstract easy to read and understand, there are no suggestions for improvement.

Introduction: authors do provide adequate background on the topic and reason for this article and describe what the authors hoped to achieve.

Results: the results presented clearly, the authors provide accurate research results, there is sufficient evidence for each result.

Conclusion: in general: Good and the research provides ample data for the authors to make their conclusion.

Tables:Need Reediting

Grammar: Need Some revision.

Author Response

We thank this reviewer for their comments.

In line with the comments from this reviewer and others, we have now made substantial changes to the tables and removed some of the fields so that they are shorter and easier to read. For example, we have merged information from table 2 and 3 and also moved some of the information to an appendix. We have simplified table 4 (now table 3) and referred readers to an appendix for more detail.  We have re-edited some of the tables.

We have made changes to the grammar as suggested.

Reviewer 2 Report

The authors have reviewed the randomized clinical trials that are being done to prevent obesity in infancy. The paper is not adding any benefit to scientific community as there is no point in reviewing the work that is going on. It is not clear what the authors want to say to the scientific world. It would have made sense if they could have concentrated on the outcome of the study, which is not possible in this case as the trials are under investigation except for 11 out of the 29 that has been published. There are many trials going on for many studies. There is no point in doing a study on the characteristics on what these trials are based on. And unless it is published, the results of the study cannot be validated. So, there is no merit to what the authors have done. The intervention components that are being used in the clinical trials may work or may not work. Based on this a review cannot be done. So if the authors have to do a review they have to wait until the results are published.

Author Response

We respectfully disagree with this reviewer's assertion that this paper brings no benefits for the scientific community. Our purpose was to provide a synthesis of new work being undertaken in the field of early prevention of obesity. Our review shows the breadth of work that is being undertaken in the field and our main aims were to synthesise the key characteristics and features of trials including important components such as behavioural targets, delivery mechanisms and agents, duration of interventions, with the view to inform future studies.

We do not agree with the premise that there is no point in doing reviewing the characteristics and interventions being conducted. Indeed, our review highlights that complex methods will need to be employed to show which intervention components are successful and at which doses and via which delivery methods. In our synthesis we highlight the necessity of describing the detail of interventions so that it is easier to deconstruct interventions to their smallest common elements in order to determine which components are actually driving the effects. We also recommend various approaches to improve reporting including standardisation of outcomes/ development of a core outcome set to facilitate harmonisation and highlight critical knowledge gaps with respect to the ideal duration of interventions and exact points at which to intervene.

As pointed out in our discussion it may be a number of years before a sufficient number of trials will be completed and their data published. This important preliminary work can be used now to inform future trials. Our publication also provides a springboard for encouraging collaboration between researchers who are working in this important area.

We note that the other four reviewers have given favourable and constructive feedback.

Reviewer 3 Report

Here the authors have compiled a detailed review of registered randomized controlled trials for the prevention of obesity in infancy. The manuscript is well written, informative and I have no concerns.

My criticisms are the tables are very lengthy and could be exhaustive for readers. Would definitely recommend to shorten them. Also, the authors can merge tables 2 and 3 into one table.  The intervention delivery agent to table 2 to and just the pertinent ones can be listed for each study

Author Response

Many thanks for this constructive feedback and we agree that the tables are lengthy. We have made substantial changes to the tables based on your recommendations and those of other reviewers. For example, we have merged information from table 2 and 3 and also moved some of the information to an appendix. We have simplified table 4 (now table 3) and referred readers to an appendix for more detail.  We have re-edited some of the tables.

Reviewer 4 Report

This paper summarizes the registered RCTs, which aimed to reduce overweight and obesity in early childhood, and gives an overview of the key features and findings from the interventions. It also points out some crucial improvements that can be useful, such as the need of standardisation in several aspects for better research designing of future interventions.

It is very well written and is well balanced between the different topics to be examined in the health promotion interventions, gathering research data that are dispersed in a paramount of research papers, and enabling to confront them. It definitely deserves to be published and is a valuable contribution to the health promotion research field.

The only weakness of the paper is its length due to tables. Thus, my recommendation/suggestion is whether is possible to convert Tables 2,3 and 4 in flowcharts/figures to show more clearly the interventions and its characteristics.

Minor comments:

Lines 42-45: I would suggest to merge these two sentences in one.

Line 426-427: ‘This review shows the breadth of work that is occurring across trials for the prevention of obesity in children, globally.’ I would suggest to be more specific in the conclusion section, adding: prevention of obesity in early childhood.

Finally, the list of references is numbered two times.

Author Response

Many thanks for this constructive feedback. We agree that the tables are lengthy and we have made some changes to these including moving some to the appendix. We have also made small changes to the other tables and removed some of the fields so that they are shorter and easier to read. 

We have also made the minor changes to the text as suggested and removed the second numbering of references.

Reviewer 5 Report

This is a very good, well-planned and executed review of ControlledTrials for the Prevention of Obesity in Infancy. The methodology of the review was well established and implemented. My only doubt is the presentation of the results with an enormous amount of data that is easy to get lost in. Is there any way to simplify the presentation of them. In the paper version it would be as many as 61 pages in its current form. The discussion was carried out in the classic way - correct.

Author Response

Many thanks to this reviewer for your comments and the suggestions for improvements. We agree that it is difficult to synthesise these data and have tried to do this in a more simplified way. For example, we have merged information from table 2 and 3 and also moved some of the information to an appendix. We have simplified the content of table 4 (now table 3) and referred readers to an appendix for more detail. The paper now is substantially shorter.

Round 2

Reviewer 2 Report

The comments that the authors have given are still not satisfactory to address the queries

Author Response

We have responded previously to this reviewers comments and respectfully disagree with them. We note that the other four reviewers have given favourable and constructive feedback and we have incorporated their helpful changes. We also note the Academic Editors response to this reviewers comments. Thank you.